# Sex Identification of Feather Color in Geese and the Expression of Melanin in Embryonic Dorsal Skin Feather Follicles

**DOI:** 10.3390/ani12111427

**Published:** 2022-05-31

**Authors:** Xiaohui Xu, Sihui Wang, Ziqiang Feng, Yupu Song, Yuxuan Zhou, Ichraf Mabrouk, Heng Cao, Xiangman Hu, Haojia Li, Yongfeng Sun

**Affiliations:** 1College of Animal Science and Technology, Jilin Agricultural University, Changchun 130118, China; xxh1417149363@163.com (X.X.); tanyaw1101@outlook.com (S.W.); fzq1031264268@163.com (Z.F.); syphandsome@163.com (Y.S.); 20211701@mails.jlau.edu.cn (Y.Z.); ichrafmabrouk189@gmail.com (I.M.); ch07253110@163.com (H.C.); hu2359258597@163.com (X.H.); jia228330@163.com (H.L.); 2Key Laboratory of Animal Production, Product Quality and Security, Jilin Agricultural University, Ministry of Education, Changchun 130118, China

**Keywords:** Holdobaggy goose, feather follicles development, melanin, feather color, sex determination

## Abstract

**Simple Summary:**

Plumage color is one of the economic characteristics of poultry. The most common plumage colors in adult geese are white and grey, so research on goose plumage color is relatively scarce compared with the molecular basis of research on chicken plumage color inheritance. Nevertheless, whether the formation of a black phenotype on the Holdobaggy geese newborn dorsal plumage color is associated with gender needs further profound study. Our study researched the gender identification of the newborn Holdobaggy geese based on plumage color and the expression levels of relative genes for melanin, which offers a theoretical basis for gender identification.

**Abstract:**

In production practice, we have found that the gray and black down on the backs of the Holdobaggy goslings is usually darker in females than in males. Melanin is the key pigment affecting the color of poultry plumage. Therefore, to determine whether the darkness of the dorsal plumage of the Holdobaggy goslings is related to sex, we study the melanin in the feather follicles of the dorsal skin during the embryonic period. The feather follicle structure and melanin distribution on the dorsal surface of the goose embryo is observed by HE staining and melanin-specific staining. The melanin content in the feather follicles of the dorsal skin of goslings is determined by ELISA. The results showed that the melanin content is higher in female geese than in males (*p* < 0.05). In addition, we also analyze the mRNA and protein expression levels of melanin-related genes (*TYRP1* and *ASIP*) by quantitative real-time PCR and Western blotting analysis. The results show that the mRNA expression level of *TYRP1* is significantly higher in the females’ dorsal skin feather follicles (*p* < 0.05), while the mRNA expression level of *ASIP* is significantly higher in the dorsal skin feather follicles of male geese (*p* < 0.05). In conclusion, the difference between males and females in the color of the black feathers on the dorsal track of the Holdobaggy goslings is verified, and it is feasible to identify the sex by the initial plumage color.

## 1. Introduction

China has a long history of animal husbandry, a diverse geographical environment, and the largest and richest goose breeding resources [1]. However, domestic geese originated from wild geese. Over thousands of years of domestication, domestic geese have diverged considerably by natural and artificial selection [2]. Among the 30 country-recognized domestic goose breeds, the Yili goose originated from the Grey goose (*Anser Anser*), while other domestic goose breeds originated from the Swan goose (*Anser Cygnoides*) [3]. In recent years, the Holdobaggy geese has been introduced and bred in large numbers in China. As the Holdobaggy geese is derived from *Anser Anser*, and they still maintain a few wild characteristics; they have strong cold resistance, and their whole body is covered with a thick layer of feathers [4]. Therefore, they have high feather production, high content, large velvet flowers, pure white velvet color, fewer impurities, good hand feeling, high lightness, and high economic value in production. The research on its germplasm characteristics has great significance for the development of animal husbandry in China.

In long-term production practice, it has been found that goose species, such as the Zhedong White geese, have yellow feathers at birth, and geese originally derived from *Anser Anser*, such as the Holdobaggy geese, have gray-black villus on the back of the goslings and on the heads of some goslings; however, the depth of this gray-black villus is related to the sex, and it gradually becomes white at 2–6 weeks of age (Figure 1). Therefore, it remains to be studied whether the dorsal plumage depth of the Holdobaggy geese is related to sex. Due to the significant difference in production efficiency of different genders, female geese have high performance in meat production, egg production, and feather production, while male geese can be used as breeding geese with higher reproductive performance. So early identification of males and females has great significance for production [5]. Consequently, if the identification of the gender can be performed in the early growth stage of Holdobaggy geese by feather color and the goslings of different genders are bred rationally, that can reduce the waste of resources, reduce the cost, improve the production efficiency, and thus improve the economic benefit.

Bird feathers are affected by many complex factors, including feather structure [6], pigment deposition [7], some endogenous factors [8], and so on. Pigment-based coloring mainly comes from pigments such as melanin, carotenoids, porphyrin, and parrot flavin. Melanin can be synthesized directly in poultry individuals, but carotenoids need to be obtained by nutrition at a late stage, so the research on bird plumage traits and related regulatory mechanisms mainly focuses on melanin. Melanins are produced by melanocytes, which are the main determinant of birds’ skin and feather color. However, melanin in poultry can be divided into eumelanin (black) and pheomelanin (yellow to orange) [9]. Furthermore, the variation of colors from white to dark black is related to the transportation of melanin particles outside the feather follicle and their deposition in the feathers [10]. The expression of melanins can be modulated by many factors. Subsequently, numerous functional genes determining feather color have been identified in poultry. Most of these genes regulate the color of feathers by affecting melanocyte migration or melanin biosynthesis. Currently, feather color-related genes are mainly melanocortin 1 receptor (*MC1R*), agouti signaling protein gene (*ASIP*), melanophilin (*MLPH*), solute carrier family (*SLC24A5*, *SLC45A2*), and tyrosinase protein families (*TYR*, *TYRP1*, *TYRP2*), etc. [11]. However, few reports have revealed the genetic mechanism of melanin-based sex determination in avian species.

Therefore, this research aims to study the distribution of melanin deposition and the expression of melanin *TYRP1* and *ASIP* genes during the goose embryonic period, thus providing a theoretical basis for Holdobaggy geese primary feather color sex determination.

## 2. Materials and Methods

### 2.1. Experimental Animals and Sampling

Holdobaggy geese embryos and goslings (at 1–2 days of age) were the subjects of this experiment. A total of 150 fertilized eggs of Holdobaggy geese were obtained from Jilin Agricultural University (Changchun, Jilin, Northeast China). The eggs were selected for the same batch of incubation and incubated in an incubator with a forced draft fan at a suitable temperature and humidity. Skin samples from the dorsum of goose embryos were sampled at embryonic day 14 (E14), the primordial period of primary feather follicles; embryonic day 18 (E18), the primordial period of secondary feather follicles; embryonic day 28 (E28), the greater developmental period of secondary feather follicles; were used for experimental analysis. The remaining samples from three stages were immediately frozen in liquid nitrogen and then stored at −80 °C in a refrigerator. The tissue block of goose embryo’s back was removed and fixed with 4% paraformaldehyde and Bouin.

### 2.2. Experiment Methods

#### 2.2.1. Gender Identification of Whole Blood

Goose embryo blood samples were collected with medical 20 μL-blood collection tubes and put into centrifugal tubes containing anticoagulant (the ratio of blood to anticoagulant was 6:1). The whole blood PCR was traced immediately after blood sampling (Table 1), and the product was determined by 1.5% agarose gel electrophoresis. The technology is referenced in Gong Daoqing’s 2013 patent.

#### 2.2.2. Preparation of Paraffin Sections

After the dorsal tissue samples were removed, the blood and stains were washed with PBS and put into a fixed fluid for fixation. The fixed tissue samples were rinsed with running water for 16 h and were dehydrated in different gradients of ethanol. Then, the tissues were soaked in xylene and treated with low and high melting point wax. Finally, sections were embedded in paraffin according to conventional methods, sectioned, and baked at 65 °C.

#### 2.2.3. HE Staining

The hematoxylin solution was used to stain the sections for 4 min, following the treatment with ethanol hydrochloride differentiation solution for 10 s and then the slides were washed with distilled water. Consequently, Eosin solution (for 4 min) and a grades series of ammoniac (for 30 s) were used to wash out the excess dye and for onwards dehydration. Subsequently, each slide was treated with xylene and sealed with neutral gum sealant.

#### 2.2.4. Silver Staining with Masson-Fontana

A freshly mixed Ammoniacal Silver Solution was placed in a 56 ℃ water bath, following the slides were incubated in the warmed Ammoniacal Silver Solution for 1 h or 12 h, and then they were washed with distilled water. Consequently, each slide was incubated in a 0.1 M Sodium Thiosulfate Solution for 3 min, washed with distilled water, and then incubated in a Nuclear Fast Red Solution for 5 min, and rinsed again with distilled water. Subsequently, the sections were dehydrated in absolute alcohol, treated with xylene, and sealed with neutral gum sealant.

#### 2.2.5. Nile Blue Staining

The slides were immersed for 20 min in a Lillie Nile Blue staining solution and rinsed with distilled water. Afterward, the samples were sealed with a water-based sealing solution (e.g., glycerol gelatin).

#### 2.2.6. Total RNA Extraction, Reverse Transcription, and Quantitative PCR

The quantitative real-time PCR analysis was used to determine the mRNA expression levels of *ASIP* and *TYRP1* genes in the embryonic goose skin with feather follicles according to different embryonic ages. The dorsal skin tissue total RNA of each sample was extracted using the Trizol Reagent (Invitrogen Life Technologies, Carlsbad, CA, USA) following the instructions of the manufacturer. The complementary DNA (cDNA) was synthesized with the RNA template by the MonScript™ RTIII Super Mix with dsDNase (Two-Step) (Monad Biotech Co., Ltd., Suzhou, China) in a total volume of 40 μL. Quantitative real-time polymerase chain reaction (qPCR) was performed using the MonAmpTMChemoHS qPCR Mix Kit (Mona Biotechnology Co., Ltd., Fuzhou, China). Each sample was replicated three times.

#### 2.2.7. Primer DESIGN

The microvolume whole blood PCR primers and real-time fluorescent quantitative PCR primers were designed according to the relevant gene sequences in the NCBI (http://www.ncbi.nlm.nih. Government) gene library by NCBI’s primer blast tool. The primer sequences are listed in Table 2. The PCR kit and Applied Biosystems provided by Biochemical Bioengineering (Shanghai) Co., Ltd. were used to carry out the PCR amplification test.

#### 2.2.8. Enzyme-Linked Immunosorbent Assay (ELISA)

Quantification of melanin content in skin follicles was performed using ELISA. The dorsal skin samples of Holdobaggy geese were prepared into tissue homogenates by adding appropriate amounts of 0.15 M sodium hydroxide solution. The melanin content of the samples was determined using a goose melanin enzyme-linked immunoassay kit (Shanghai Bohu Biotechnology Co., Ltd., Shanghai, China). Samples are added in triplicate.

#### 2.2.9. Protein Extraction and Protein Degeneration

The total protein samples for electrophoresis were extracted using radio immunoprecipitation assay lysis buffer (including protease and phosphatase inhibitors). Protein concentration was determined by diluting the total protein solution 20-fold using the BCA Protein Quantification Assay Kit (Thermo Fisher Scientific Co., Ltd., Shanghai, China). Three replicates were set up for both standard curve wells and sample wells. After diluting the total protein solution with lysate for a uniform concentration, protein denaturation was performed by adding a 5× protein loading buffer.

#### 2.2.10. Western Blotting Analysis

Western blotting was used to quantify the TYRP1-related protein expression in dorsal skin samples between groups. The denatured proteins were subjected to electrophoretic gel running. For each sample, the volume containing 10 μg total proteins was pipetted onto a 4–20% SDS-PAGE and Tris-Glycine SDS electrophoresis buffer for 1 h. The transferred proteins were bound to the surface of the PVDF membrane for 30 min. The membranes were blocked with 5% skim milk powder for 5 h at 4 °C and then incubated with the primary antibody (Rabbit IgG, Chengdu Zhengneng Biotechnology Co., Ltd., China) at 4 °C for 16 h. In the end, the membranes were incubated with a Goat Anti-Rabbit IgG (H + L)/HRP antibody secondary antibody (Bioworld Technology, Inc., St. Louis Park, MN, USA) for 2 h at room temperature. The membranes were visualized with an ECL Test Kit (Millipore, Darmstadt, Germany).

### 2.3. Feather Follicles Morphology and Structure Observation

Tissue sections were observed with an Olympus Corporation light microscope (eyepiece 10, Olympus, Tokyo, Japan). Sections were first observed using a 4× low objective, then sequentially using 10× and 20× objectives, and typical sections were photographed.

### 2.4. Statistical Analysis

The relative gene expression was calculated by the Warp 2^−ΔΔCT^ method, and the expression of each protein was obtained by measuring the grayscale value by ImageJ software and comparing the grayscale value of the target protein with that of the internal reference protein. All data distribution was analyzed using the D’Agostino–Pearson test. The results were expressed as mean ± SEM statistical significance was determined using one-way ANOVA by Tukey test as a post hoc to evaluate the parameters recorded over time for each group and to evaluate differences among groups. The significant difference between the data was considered as *p* < 0.05.

### 2.5. Ethics Statement

This experiment was approved by the Animal Ethical and Welfare Committee of Jilin Agricultural University (Approval No. GR (J) 18-003).

## 3. Results

### 3.1. Microvolume Whole Blood PCR for Sex Determination

The PCR results are consistent with the expected target band size. As shown in Figure 2, lanes 1, 4, 5, 7, and 8 are single bands, about 485 bp in size, which can be judged as male geese embryos, and lanes 2, 3, 6, 9, and 10 are double bands, sized about 330 bp and 485 bp, which can be judged as female geese embryos.

### 3.2. Dorsal Skin Feather Follicle Development and Melanin Distribution in Feather Follicles during Embryonic Period 

#### 3.2.1. Visual Observation of Dorsal Skin Feather Development of Geese Embryos

The physical observation of embryos’ dorsal skin feathers was carried out. As shown in Figure 3, the dorsal skin of E13 geese embryos presented uniformly elevated and well-arranged feather buds, in which the female embryos appeared partially black on the dorsal feather buds, while the dorsal feather buds of the male embryos did not show any obvious black.

The dorsal feathers of geese embryos at E18 were keratinized. The feathers on the backs of male and female geese had a large area of obvious black, which started from the bottom of the feathers. The dorsal skins of geese embryos at E28 were completely covered by black and yellow feathers, and the dorsal feathers of male and female geese were lighter than those at E18.

#### 3.2.2. Histological Structure of Feather Follicles at Different Stages

The histological sections examining skin morphogenesis at three embryonic developmental stages (E13, E18, and E28) are shown in Figure 4. The results indicate that at the early embryonic stage of E13, short feather buds were clearly distinguishable in the feather tracts, which had formed by cell proliferation in the epithelium (Figure 4A). By E18, the basal skin epidermis of the plumule was invaginated, and the distal end of the plumule began to grow (Figure 4B). The diameter of secondary feather follicles was smaller than that of primary feather follicles, which was in the stage of secondary feather follicle development. At E28, the degree of epidermis invagination was deeper, and during this period, the muscles, nerves, glands, and blood vessels connected with mature feather follicles gradually became abundant in the dermis. The dermal papilla in the ellipsoid shape was observed at the end of the feather follicles, which was in the period of the large development of secondary feather follicles in the embryonic stage (Figure 4C).

With the Fontana-Masson silver staining method, melanin granules (melanocytes and argyrophilic cells) were black, and other cells appeared pink through the reduction of melanin. Therefore, the melanin and lipofuscin were black, and other cells were blue or uncolored with Nile blue staining. By combining the two specific stains for melanin, the distribution of melanin in skin feather follicles could be clearly observed. The staining results showed that melanin was mainly distributed on the surface of the feather buds at E13, with small amounts of melanin deposited in female geese, while there was no obvious distribution of melanin in male geese (Figure 5).

### 3.3. Melanin Content in Embryonic Dorsal Skin Feather Follicles of Holdobaggy Geese

The melanin content of Holdobaggy geese at three time points during the embryonic stage first increased and then decreased slightly, and the highest content was observed at E18 and the lowest at E13. There were significant differences among female geese at E13, E18, and E28 (*p* < 0.05), while no differences were observed between E18 and E28 (*p* > 0.05), and the melanin content showed extremely significant differences among male geese at E13, E18, and E28 (*p* < 0.01) (Figure 6).

Furthermore, the melanin content significantly differed between male and female geese in E13 and E18 (*p* < 0.05), but no differences were noted between E18 and E28 (*p* > 0.05), with the highest melanin content observed in females than in males at each time point. Combined with the previous visual observations of dorsal feather color and staining of dorsal tissues at these three time points, melanin deposition followed the same trend as melanin content (Figure 6).

### 3.4. The mRNA Expression Levels of ASIP and TYRP1 Genes in the Embryonic Holdobaggy Geese Skin Feather Follicles

The gene expression of male and female Holdobaggy geese at three time points of the embryonic stage had different trends. The mRNA expression of the *TYRP1* gene for female geese showed an up-regulated pattern in the middle stages of embryonic development (E13 and E18) and decreased slightly in the later stages. Moreover, *TYRP1* expression was highest in E18, and the lowest expression was shown in E28, from which there was a difference between E18 and the other two time points (*p* < 0.05). However, there was no significant difference in the relative expression of *TYRP1* gene mRNA levels in E13 and E28 (*p* > 0.05). From the entire developmental stages of skin with feather follicles, *TYRP1* gene mRNA of male geese showed a downward trend from E13 to E28, in which there was no difference in *TYRP1* mRNA relative expression between E13 and E18 (*p* > 0.05), and there was a difference between E28 and the other two time points (*p* < 0.05), with the highest expression being observed at E18 and the lowest at E28. On the other hand, the mRNA relative expression was not significantly different between female and male geese in E13 (*p* > 0.05), but the expression of the *TYRP1* gene was higher in female geese than in male geese at E18 (*p* < 0.05) (Figure 7).

Furthermore, the mRNA of the *ASIP* gene in female geese was highly expressed at E28, increased at E13, and decreased at E18, which formed a V-shape showing a no significant difference in the relative expression of *ASIP* gene mRNA between E13 and E28 (*p* > 0.05), and an extremely significant difference between E18 and E28 (*p* < 0.01), as well as there was no difference between E13 and both E18 and E28 (*p* > 0.05). However, the highest expression was at E28 and the lowest was at E18. However, the mRNA expression trend of the *ASIP* gene in male geese was completely the opposite and increased gradually from E13 to E28. There was a significant difference between each of the three time points (*p* < 0.05), with the highest expression shown at E28 and the lowest at E13. For the relative mRNA expression between female geese and male geese at E13 showed no significant difference (*p* > 0.05), but there were significant differences in the relative mRNA expression between female and male geese at E18 and E28 (*p* < 0.05), and the expression in both female geese was lower than that in male geese (Figure 7).

### 3.5. Protein Expression Levels of TYRP1 in the Embryonic Dorsal Skin Feather Follicles of Holdobaggy Geese

The Western blot results confirmed that the expression level of TYRP1 protein at E18 was significantly different from E13 and E28 at all three points in the embryos of the Holdobaggy geese (*p* < 0.01). Only E18 was expressed, and the expression in female geese was higher than in male geese at this time point (*p* < 0.01) (Figure 8).

### 3.6. The Sex Differences in Postnatal Dorsal Coat Coloration in Holdobaggy Geese

In total, 2000 newborns (1–2 days old) of Holdobaggy geese were randomly compared in terms of back color shade to distinguish males from females. We found that those with darker backs were female Holdobaggy geese, while those with lighter backs were male Holdobaggy geese (Figure 1A). The success rate of sex identification was as high as 99% for the goslings that were distinguished as male and female by traditional anus-turning.

## 4. Discussion

In this study, we observed the distribution of feather follicles and melanin in the embryonic dorsal skin of the Holdobaggy geese by HE staining and specific staining for melanin, and we found that the melanin is responsible for the black color of the dorsal plumage. Moreover, melanin was mainly in the feather follicles and was deposited in large quantities on the top of the feather follicles and in the growing feathers. The different colors of feathers are related to the amount, characteristics, distribution, and form of melanin [12]. Similar to our findings, as shown in Figure 4, where melanin was deposited more in the skin follicles of female geese compared to male geese at three time points during the embryonic stage of the Holdobaggy geese, some studies have demonstrated that the number of melanin granules in white hairs or feathers is limited [13]. The quantitative analysis of melanin content in the embryonic dorsal skin follicles of the Holdobaggy geese also showed that the females had more melanin compared to the males (*p* < 0.05) during embryonic development, and the high deposition of melanin particles in the feather follicles directly contributed to the darker plumage color.

*TYRP1* is the first successfully cloned pigment gene, which encodes a protein homologous to tyrosinase and is localized at the Brown site in mice [14]. This gene is an important downstream target gene in the regulation of melanin synthesis by affecting melanosome maturation, melanin synthesis process, and *TYR* activity [15]. In the process of melanin synthesis, high, and low *TYRP1* activity can lead to the synthesis and conversion of eumelanin and pheomelanin. However, high *TYRP1* activity leads to the production of eumelanin, while low *TYRP1* activity leads to the production of pheomelanin [16]. Furthermore, a study demonstrated that the mutation of the chicken *TYRP1* gene significantly affected the skin color, tibia color, and claw color of silk-feathered black chickens, and the expression of the *TYRP1* gene was consistent with the deposition and distribution of melanin [17]. In contrast, Gratten et al. found that there was no significant difference in *TYRP1* gene expression between dark and light-colored Soai sheep [18]. Similarly, data from a study on the *TYRP1* gene in Korean quails revealed that the expression level of the *TYRP1* gene was not consistent with the color depth of Korean quails, and the higher the expression level of the *TYRP1* gene was, the lighter the color of Korean quails [19]. This suggests that the *TYRP1* gene may not be the main control gene for melanin formation in Soai sheep and Korean quail. However, studies have shown that the expression of the *TYRP1* gene is higher in black ducks than in White ducks [20]. Other studies have found that the *T**YRP1* gene expression level in brown feather sacs was the highest among the three species of spotted-billed mallard but decreased in yellow and white feathers, but not in white feathers [21]. Therefore, it is speculated that the *TYRP1* is expressed differently in the dorsal skin feather follicles of geese embryos with different dorsal plumage. Our results support this hypothesis. The mRNA and protein expression of the *TYRP1* gene were highest in the skin feather follicles of E18 geese. In summary, *TYRP1* is the main control gene of melanin biosynthesis in the embryonic period of the Holdobaggy geese, and the higher the expression level of *TYRP1*, the more melanin synthesis and deposition, resulting in darker plumage.

*ASIP*, a candidate master effector gene for the feather color melanin trait, is present in almost all vertebrates. In addition, *ASIP* is secreted by dermal papilla cells closed with melanocytes and acts only in the feather follicle microenvironment, regulating the melanin synthesis in time and space [22]. The expression of the *ASIP* gene causes pheomelanin expression, while *ASIP* gene non-expression causes eumelanin expression [23]. Several studies have argued that the expression of *ASIP* in white goat skin tissues was more important than in black goat skin tissues, with a significant difference [24,25]. Moreover, it has been demonstrated that mutations and abnormal expression levels of the *ASIP* gene can cause changes in feather color in birds [26]. Hiragaki et al. (2008) [27] analyzed the expression of the *ASIP* gene in four plumage colors (recessive black plumage, wild chestnut plumage, camel-colored plumage, and yellow plumage) of Japanese quail by real-time quantitative fluorescence PCR, and found that the expression of the *ASIP* gene in recessive black plumage was significantly lower than that of the *ASIP* gene in wild chestnut plumage (*p* < 0.05). Thus, Zhang et al. (2005) [28] analyzed the expression quantity of the *ASIP* gene in three plumage colors (black, chestnut, and white) of Korean quail. The results showed that the expression of the *ASIP* gene in black plumage was significantly lower than that in white plumage. On the other hand, numerous studies have argued that the relative expression of the *ASIP* gene in each of the tissues of the muscle, kidney, liver, myogastric, and skin of healthy white velvet black bone chickens was in the opposite order of melanin content [29]. Mikus Abolins-Abols et al. (2018) [30] illustrated that the high expression of the *ASIP* gene was in grey-headed feathers compared to black-feathered rushes finches, suggesting that *ASIP* may decrease melanin production in these feathers. In this study, the mRNA expression of the *ASIP* gene was lower in the cutaneous feather follicles of the female than in the male Holdobaggy geese in E18 and E28. It decreased and then increased at three time points during the embryonic stage in female geese, with low expression in E18, while it continued to increase in the feather follicles of the skin of Holdobaggy male geese. The high expression of *ASIP* in male geese embryos may inhibit the formation of melanin. Whereas whether the expression of the *ASIP* gene at the protein level is varied remains to be further studied.

In addition, the mRNA and protein levels of TYRP1 were inconsistent in this study. We speculated that this situation may be related to genetic differences. According to the central dogma, DNA undergoes a series of transcription and translation in the process of forming proteins, and the transmission of genetic information in the process of genetic information that seems simple but very complex may be due to its own reasons or affected by uncertainties, resulting in deviations in the expression of mRNA levels and protein levels.

At present, the methods of sex identification of poultry mainly include turning the anus, distinguishing fast and slow feathers, and the identification method of feather color. Among the above methods, the identification method of turning the anus has a wide range of applications and high accuracy, but the method has limitations and will affect the performance of the poultry in the later stages. However, feather sex identification has the characteristics of being easy to grasp, low cost, high efficiency, time-saving, etc., and it does not cause harm to individual poultry, which is increasingly applied to the early sex identification of poultry. In terms of genetics, parts of the progeny’s traits are controlled by genes on the sex chromosomes, and their phenotypes vary by sex difference. According to this characteristic, the difference in plumage color can be used to identify males and females in poultry production. It has been widely studied and applied in poultry such as chickens [31], ducks [32], and quails [33]. Studies on goose feathers have focused on adult goose species domesticated from *Anser Anser*, while there have been few studies on the feather color of goslings. In this study, the difference between males and females in the color of the black feathers on the back of the Holdobaggy geese was verified by experiments at the embryonic stage and the postnatal period. Those with darker backs were female Holdobaggy geese, while those with lighter backs were male Holdobaggy geese. It is feasible to identify the sex of the Holdobaggy geese from the initial plumage color.

## 5. Conclusions

In production practice, early sex identification can be performed on the goose species domesticated from *Anser Anser* by the depth of the black feathers on the back of which the black feathers on the back of the goslings are deeper for females and shallower for males. This method reduces production costs and improves production efficiency, which can be effectively popularized and used in production.

## Figures and Tables

**Figure 1 animals-12-01427-f001:**
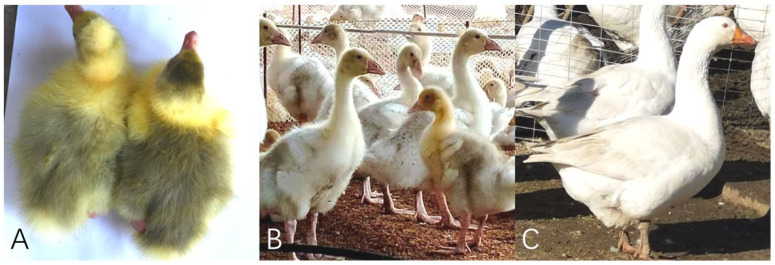
Color change process of Holdobaggy geese in different growth stages. Note that (**A**) is the gosling’s newborn (1–2 days old) with black and gray feathers on their backs and some heads; (**B**) is the young goslings aged 4–6 weeks in the molting stage with white feathers on their backs; (**C**) is the adult geese with white feathers on their whole body.

**Figure 2 animals-12-01427-f002:**
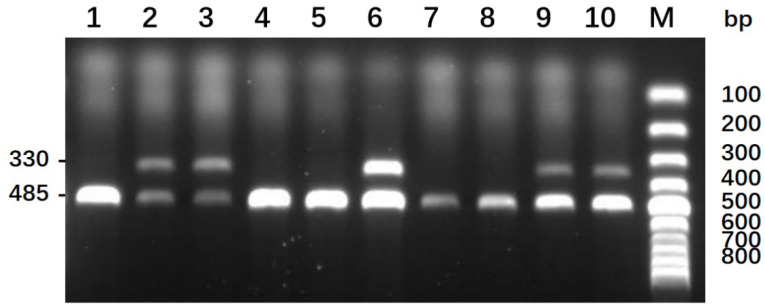
Sex identification results of micro whole blood PCR. Note: double bands are female geese (330 bp and 485 bp) and single bands are male geese (485 bp).

**Figure 3 animals-12-01427-f003:**
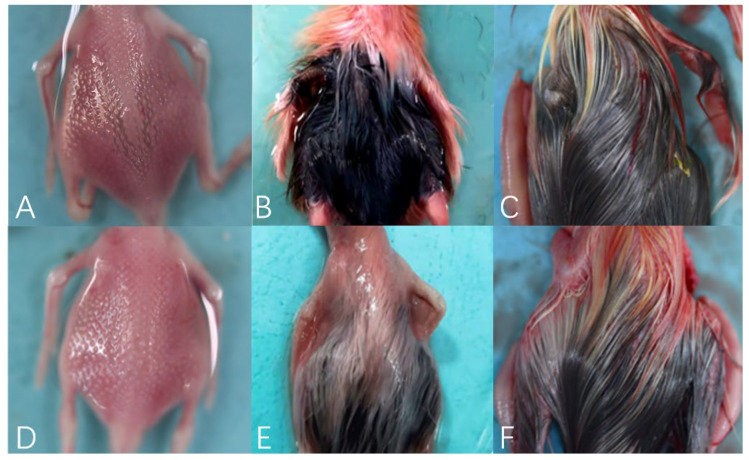
Visual observation of dorsal skin feathers development of Holdobaggy geese during embryonic stage. Note: (**A**–**C**). Visual observation of dorsal skin feathers development of female Holdobaggy geese embryos at E13, E18, and E28, respectively, and (**D**–**F**) Visual observation of dorsal skin feathers development of male Holdobaggy geese embryos at E13, E18, and E28, respectively.

**Figure 4 animals-12-01427-f004:**
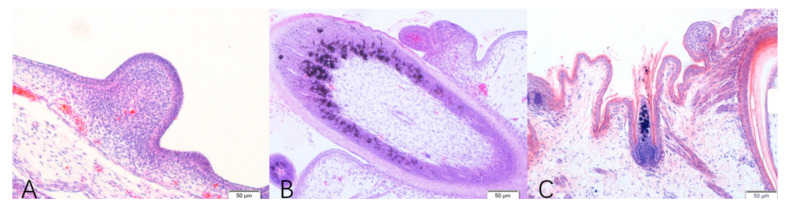
Microscopic observation of geese embryo skin at three different developmental stages by HE. Note: (**A**) primordial period of primary feather follicles at embryonic day 13, (**B**) primordial period of secondary feather follicles at embryonic day 18, and (**C**) greater developmental period of secondary feather follicles at embryonic day 28. Magnified: 10×; Bar: 100 μm.

**Figure 5 animals-12-01427-f005:**
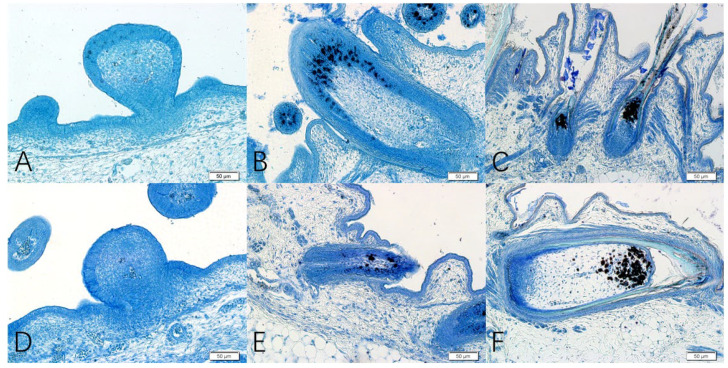
Results of Nile blue staining of Skin and Feather Follicles in geese male and female Embryo Note: (**A**–**C**) the melanin distribution at E13, E18, and E28 of the female Holdobaggy geese, respectively. (**D**–**F**) the melanin distribution at E13, E18, and E28 of the male Holdobaggy geese, respectively. Magnified: 20×; Bar: 50 μm.

**Figure 6 animals-12-01427-f006:**
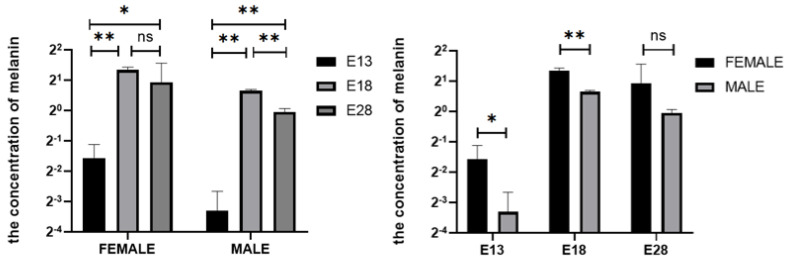
Melanin content of dorsal skin feather follicle of Holdobaggy geese during embryonic stage. Note: ns no significant difference (*p* > 0.05), * Significant difference (*p* < 0.05), ** extremely significant difference (*p* < 0.01). The results are expressed as the mean ± SEM. n:3.

**Figure 7 animals-12-01427-f007:**
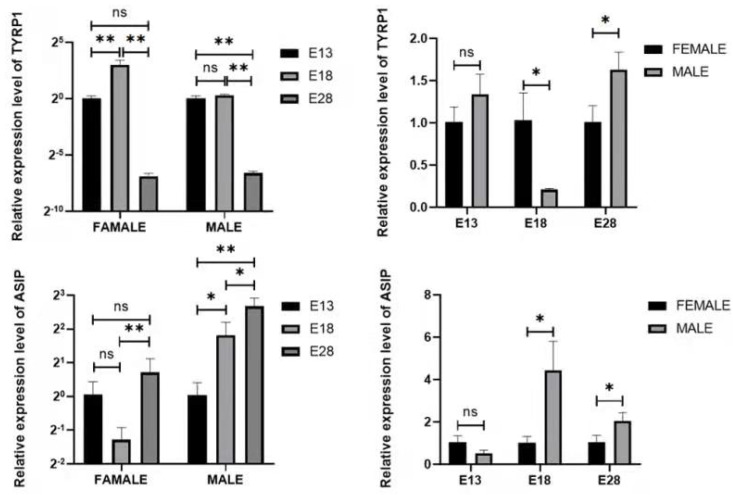
The expression of *TYRP1* and *ASIP* genes in Holdobaggy geese dorsal skin feather follicles during embryonic stage. Note: ns no significant difference (*p* > 0.05), * Significant difference (*p* < 0.05), ** extremely significant difference (*p* < 0.01). The results are expressed as the mean ± SEM. n:3.

**Figure 8 animals-12-01427-f008:**
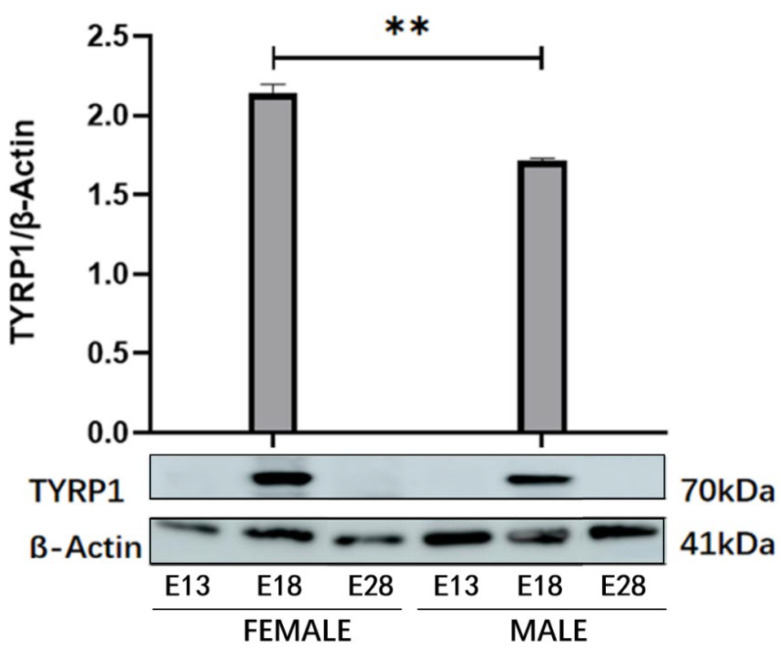
Protein expression trend of TYRP1 in the embryonic dorsal skin feather follicles of Holdobaggy geese using Western blot. Note: ** Significant difference (*p* < 0.01). Each sample is shown as mean ± SEM. The upper panel shows the measurable quantities of TYRP1 in dorsal skin feather follicles at different growth stages of male and female geese. The lower panel indicates the β-actin as an internal control in all samples. Full Western Blot Figures are provided in the Appendix A.

**Table 1 animals-12-01427-t001:** Micro-trace whole blood PCR system.

Try, the Agent	Body, Product
2× San Taq PCR Mix	25 μL
Forward Primer	2 μL
Reverse Primer	2 μL
whole blood	1 μL
ddH_2_O	Up to 50 μL

**Table 2 animals-12-01427-t002:** Micro-trace whole blood PCR with real-time PCR primer sequences.

Gene Name	Primer Sequences (5′–3′)
*CHD1*	F: GGTGGCTTAATGAGGTAGCA R: AGGATGGAAATGAGTGCA
*ASIP*	F: GCCAAATTAGCAGCACTTCC R: TTGCCACATTGCCATTCTTGG
*TYRP1*	F: CGGCAATACAACATGGTGCC R: AAGCTTTCAGGGAGGAAGACA

The candidate genes: *CHD1*: Chromodomain Helicase DNA-Binding Protein 1; *ASIP*: Agouti Signaling Protein and *TYRP1*: Tyrosinase-Related Protein 1. F denotes forward primers and R denotes reverse primers.

## Data Availability

The data presented in this study are available on request from the corresponding author.

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
