# Peer review of "Sex Identification of Feather Color in Geese and the Expression of Melanin in Embryonic Dorsal Skin Feather Follicles"

_animals, 2022, doi:10.3390/ani12111427_

Round 1

Reviewer 1 Report

Dear authors,

I read your manuscript with great interest and I found merit in idea, experimental plan and methods proposed. However, the manuscript displays some problems that prevent from publication at this stage.

The main problem is the language. The text is full of grammar errors. In any case,I am othe opinion that language shouldn't affect the evaluation of the paper as you solve it easily with professional proof check. 

The manuscript is well structured, though I would suggest to implement the following aspects:

  • melanogenesis and pigmentation in birds. There are also other factors affecting plumage  colours particularly in birds like carotenoids (xantins and carotenes) and tyrosine with some microelements like copper for instance. I believe that this is worthy of being stated in the introduction, and I would suggest you to refer to Cappai et al., 2017, Ecol Evo 7:390-398. and Cappai et al.,2015 It J Anim Sci, 14:50502-507
  • Methods are appropriate and innovative.results are clearly displayed and informative 
  • Discussion should better focus on the practical application of your result. Can you forecast the effects of your findings in future management ?
  • Conclusion must be rewritten in full. They cannot report short tracts of the abstract. They must focus on the take home message to the reader.

Please, check English grammar and style. 

Author Response

Thank you for your comments and suggestions on this article. For your guidance, I made the following modifications to the article.

First, I have revised and improved the full text for the problem of language grammar.

Second, I consulted the literature you recommended, which is of great reference significance. As you said, there are indeed many kinds that affect feather color, which I supplemented and improved in the preface of the article. However, for the geese and newborn goslings in the embryonic stage in this study, the main influence on the color is melanin, because the early embryonic development stage does not involve eating problems and the influence of too many external factors.
Third, the research has important practical significance. In the post natal experiment of this study, we randomly classified 2000 goslings hatched in the same batch and divided them into deep black feather group (A) and light black feather group (B) according to the depth of black feathers on the back. Then we used the traditional anal turning method to identify the gender of the two groups. The rate of female goslings in group A and male goslings in group B reached more than 99%, which showed that it was feasible to use feather color for gender identification, The experiment in embryonic stage provides a useful theoretical basis for this method. In view of this problem, I improved the discussion part.
Fourth, the conclusion part has also been modified according to your suggestions, so that readers can quickly obtain useful information.

Thank you again for your comments.

Reviewer 2 Report

The manuscript offers the information about the differences in melanin contents and related gene expression in the dorsal skin of goslings between sexes. The methods used in the study are acceptable, and the results are interesting. However, the writing skill is expected to be improved. Some sentences in “RESULTS” should be described in “INTRODUCTION”, not in “RESULTS”, e.g., Lines 191-195. Some sentences in “RESULTS” should be moved to “MATERIALS AND METHODS”, e.g., Lines 315-319. Some sentences in “DISCUSSION” repeatedly describe methods, e.g., Lines 388-391. The citations in the context have many errors or omission, e.g., Line 362, Line 372. The rule of capitalization is not consistent in the whole manuscript. A space between numbers and units (e.g., μl, h, M, etc.) does not to be omitted. There are many grammar and punctuation mark errors should be corrected and the manuscript is suggested to be improved by professional English editing. Some significant mistakes are listed below:

  1. Line 24: “However, the results” CHANGE TO “The results”.
  2. Line 25: “geese (P < 0.05)” CHANGE TO “geese than in the males (P < 0.05)”.
  3. Line 29-30: The sentence “In conclusion, ….” should be rewritten.
  4. Line 54: “too” CHANGE TO “to”.
  5. Line 79: “Tyrosinase” “tyrosinase”.
  6. Line 84: “theoretical a basis” CHANGE TO “a theoretical basis”.
  7. Line 92: “skin” CHANGE TO “Skin”.
  8. Line 99: “methods” CHANGE TO “Methods”.
  9. Line 101: “a medical 20 μL blood vessels needles” CHANGE TO “medical 20 μL- blood vessel needles” (blood vessel needles?)
  10. Line 102: “a centrifugal tubes” CHANGE TO “centrifugal tubes”.
  11. Line 103: “blood whole PCR” CHANGE TO “whole blood PCR”.
  12. Table 1: “??μl” CHANGE TO “?? μl” or “?? μL”. (Note the space and consistence of capitalization)
  13. Line 110 & 111: “ethanol” and “wax” CHANGE TO “ethanol.” & “wax.” (Note the period).
  14. Line 125: “0.1M” CHANGE TO “0.1 M” (Note the space.)
  15. Line 132: “e. g.,” CHANGE TO “e.g.,” (Note the space.)
  16. Line 139-140: “(Mona Biotechnology Co., Ltd.) Kit.” CHANGE TO “(Mona Biotechnology Co., Ltd.).” (DELETE THE WORD “Kit”.)
  17. Line 153: “Skin” CHAGE TO “skin”. (Note the lower case.)
  18. Line 161: Please describe the source of the assay kit.
  19. Line 182: “mean + SEM” CHANGE TO “mean ± SEM”.
  20. The errors in many other places should be corrected also.

Author Response

Thank you for your comments and suggestion on this article. For your guidance, I made the following modifications to the article.

First, it improves the language of the full text. Including grammar problems, letter case problems and punctuation problems in the article.
Second, some structures have been readjusted. The content structure of "result", "test method" and "conclusion" is improved. 1) The description of the test method in the results is adjusted to the method part; 2) Delete the description of the theory and method of micro whole blood sex identification in the results; 3) Delete the repetitive description of the method in the conclusion.
Thirdly, the references in the full text are re cited. Due to the increase of content, some references were added and the citation position of the discussion part was corrected.
Fourth, the specific issues you listed have been revised, and the introduction and discussion contents have been further added.

Thank you again for your comments.
